# Influence of AWJ Process Parameters on Erosion Groove Formation in Additively Manufactured Stainless Steel

**DOI:** 10.3390/ma17122964

**Published:** 2024-06-17

**Authors:** Radoslav Vandžura, Vladimír Simkulet, František Botko, Matúš Geľatko, Michal Hatala

**Affiliations:** Faculty of Manufacturing Technologies, Technical University of Košice with Seat in Prešov, 080 01 Prešov, Slovakia; vladimir.simkulet@tuke.sk (V.S.); frantisek.botko@tuke.sk (F.B.); matus.gelatko@tuke.sk (M.G.); michal.hatala@tuke.sk (M.H.)

**Keywords:** abrasive water jet, additive manufacturing, stainless steel, 316L, erosion track, green machining

## Abstract

The presented manuscript focuses on the influence of process parameters of abrasive water jet technology on the creation of non-transient erosive grooves. The processed stainless steel SS 316L is additively manufactured using the selective laser melting (SLM) method. Due to the distinct mechanical properties of this material resulting from the production process, the material was machined in two planes according to the direction of the printing layers. The experimental part employed a planned experiment utilizing the DoE (Design of Experiment) method. Experiments aimed at varying process parameters (traverse speed, standoff distance, abrasive mass flow) were conducted at a water pressure of 50 MPa, assessing the parameters’ impact on the removed material and the properties of the resulting non-transient erosion groove. The properties of the erosion groove, such as shape and the material removal (area of erosion groove), were evaluated. The influences of process parameters on the observed parameters were assessed using the analysis of variance (ANOVA) method. Experiment preparation and setup were based on a thorough theoretical analysis of the machining process with the abrasive water jet (AWJ) method. The experiment also highlights the diverse properties of the SS 316L material prepared using the SLM method when machined with AWJ technology.

## 1. Introduction

Advancement in the abrasive water jet (AWJ) machining process offers a precise and efficient method of cutting and machining difficult-to-machine metal alloys. One of the key aspects influencing the performance of AWJ machining is the process parameters, such as water pressure, traverse speed, size and mass flow of abrasive particles, and nozzle parameters [1,2,3]. The mentioned parameters directly affect the machining efficiency, surface quality, and precision of the final product. Understanding technological parameters’ interaction is essential for the optimization of machining processes and achieving the desired results [4,5,6].

AWJ can be used for various material shaping or surface modification operations, applying an appropriate combination of technological parameters. One specific operation is the removal of material from the surface of materials, similar to traditional milling [6,7]. This application of the AWJ method is particularly important for new difficult-to-machine materials such as nickel, chromium, tungsten, and titanium alloys prepared by 3D printing methods.

Selective Laser Melting (SLM) belongs to technologies where the input base material is in the form of metal powder. In the case of metallic powder materials, this input material has a composition similar to conventionally produced metals. Therefore, the chemical compositions of conventional steels and products of SLM technology should have similar mechanical and chemical properties. Variations in these properties are mainly influenced by the particular additive production method used [8,9,10]. Significant deviations in mechanical properties and preparation methods are noticeable, especially in Laser Powder Bed Fusion (L-PBF) methods where the principle of creating the finished part lies in the sintering of metal powder layer-by-layer until the desired shape of the component is achieved. The mechanical properties of such printed parts vary along the printing directions. Hence, it is important to observe the influence of technological parameters during the machining of these materials in different layering directions of printing [11,12,13].

For difficult-to-machine additively prepared materials such as nickel, chromium, and titanium alloys, the application of the AWJ method appears as a promising technology for the surface modification of printed parts. Many authors mention terms like “waterjet milling”, controlled depth of cut, and so on. One of the advantages of using an abrasive water jet for milling is the absence of a heat-affected zone, which can be appropriate for the processing of heat-sensitive materials. Additionally, the risk of tool wear and minimizing the generation of dust or vapors are other positives, due to the absence of a rigid tool [9,14,15].

The idea of milling using the AWJ (abrasive water jet) method dates back to the 1990s. The main pioneer of this method and the first to conduct and describe experiments was Hashish, Mohamed, in the book “An Investigation of Milling With Abrasive Waterjets: A Preliminary Investigation” [8]. The book showcased the primary advantages of utilizing this technology in the machining of aluminum and titanium alloys. It has been determined that the material removal rate from the surface is primarily influenced by the appropriate combinations of technological parameters: the traverse speed of the cutting head, abrasive mass flow, standoff distance, material properties, and water pressure [8,16,17]. In further works by Hashish et al., experiments describe the influences of the water pressure on the depth of cut. The water pressure varied from 120 MPa to 360 MPa in the experiments. The experiments elucidate the dependency of the resulting depth of cut on the operating water pressure [17]. In the research conducted by Alberdi et al. in 2010, it was found that the depth of cut depends on the feed rate of the cutting head. The study demonstrates the challenge of maintaining a constant depth of cut during the machining and emphasizes the importance of optimizing the entire process and individual input process parameters [6]. The influences of additional parameters such as the diameter of the machining nozzle, abrasive mass flow rate, traverse speed, and standoff distance are described in the study by Rabani et al. It was explained how each process parameter positively or negatively affects the controlled material removal process. The study demonstrates the potential for predicting and correcting the individual input parameters of the process [18]. 

AWJ milling in its simplest form consists of a single pass of the cutting head. Through predefined movements of the cutting head, it is possible to create milled pockets of complex shapes. The entire material removal process depends on the water jet pressure, the movement strategy of the cutting head, and the diameter of the nozzle [6,7,18,19]. In the milling of complex geometries, programming the motion of the cutting head involves the implementation of advanced motion strategies such as contouring, pocketing, or adaptive toolpaths. These strategies define how the cutting head moves relative to the workpiece to achieve the desired shape. Additionally, models must be developed to accurately control the depth of cut throughout the machining process. These models may incorporate feedback mechanisms, real-time monitoring, or predictive algorithms to adjust cutting parameters and ensure consistent and precise material removal. Combining the sophisticated motion programming with precise depth control models, manufacturers can effectively mill complex components with high accuracy and efficiency [7,19]. Predictive models of AWJ milling were developed by Deam et al. These models demonstrated that the depth of cut increased with increasing flow of abrasive material, traverse speed, and nozzle diameter. Additionally, they revealed that material properties and standoff distance also influenced the depth of the cut. These predictive models serve as valuable tools for optimizing the machining process and achieving the desired machining results [19]. The water pressure in the study by Escobar-Palafox et al. exhibited a nonlinear relationship concerning the depth of cut, highlighting the necessity of stable water pressure. This suggests that varying water pressure levels can have nonlinear effects on the depth of cut, indicating the importance of carefully controlling and optimizing water pressure in the machining process. Appropriate control of water pressure can significantly impact the quality, precision, and efficiency of material removal during the abrasive water jet milling. Understanding and regulating the nonlinear relationship between water pressure and depth of cut is crucial for machining with controlled depth [7]. 

The application of a low-pressure abrasive jet and monitoring of the controlled depth of cut at 50 MPa was published by Botko et al. The study highlighted the erosion effects on the Ti6Al4V alloy. It pointed out the interaction of input parameters such as traverse speed, abrasive mass flow, and tilt of the cutting head. This research clarifies the complex dynamics involved in achieving precise material removal and surface finish in the machining of challenging materials like Ti6Al4V alloy [20].

Experimental AWJ machining was applied to various types of stainless steels, low-carbon steels, tool steels [6,21], aluminum alloy AW 2024 [22], and titanium alloys [7,23].

According to the published research, there are no documented experiments on the application of AWJ (AWJ milling, controlled depth machining) on AM (Additive Manufacturing) materials. This provides an opportunity for experiments, especially for difficult-to-machine AM materials. Described experiments in the presented research focus on stainless steel alloys prepared by the SLM method and machined using low-pressure AWJ. The use of low water pressure (50 MPa) reduces the overall energy demand of the process and can be classified as a green machining process. This approach is promising for effective machining of AM materials, particularly those with challenging machinability characteristics, with minimal environmental impact.

The desired industrial application of the presented research is in the preparation of surfaces of AM parts for subsequent technological operations such as fusion welding. Surface layers of AM components exhibit different mechanical properties compared to subsurface layers.

## 2. Materials and Methods

Experiments focused on the application of abrasive water jet (AWJ) were conducted to explore the material removal and the formation of non-transient erosive grooves, while the parameters of the erosion groove during changes of process parameters were observed. The water pressure was set at 50 MPa. Material removal experiments using AWJ technology, where erosive grooves were formed with a single pass of the AWJ, were focused on the investigation of the influence of the technological parameters—traverse speed (*v_f_* [mm·min^−1^]), abrasive mass flow rate (*m_a_* [g·min^−1^]), and standoff distance (*SoD* [mm])—on the dimensions of the groove, its shape and volume, the amount of removed material, and the maximum and minimum depth. 

A selected experimental material was an additively produced material SS 316L. This additively produced material is prepared using Renishaw’s, (Renishaw, Wotton-under-Edge, UK) selective laser melting (SLM) technology from a metal powder Renishaw SS 316L-0407. The 316L-0407 alloy is an austenitic stainless steel alloyed with chromium up to 18%, nickel up to 14%, and molybdenum up to 3%, along with other alloying elements. This alloy is a low-carbon variation of the standard 316 alloy. Renishaw powder materials are supplied according to strict specifications to minimize variations between individual batches. The powder composition is shown in Table 1 [24].

Specimens of the selected material for experiments were prepared using the Renishaw AM500E 3D printer. The AM500E system (Protolab, Ostrava, Czech Republic) with a working chamber of 250 × 250 × 350 mm operates with a laser power of up to 500 W. The maximum scanning velocity is 2000 mm.s^−1^, and the layer thickness ranges from 20 µm to 100 μm. Powder material particles are deposited in the desired layer and subsequently melted and fused by a laser beam. A homogeneous layer of material forms during the cooling process in the laser melt pool. The process of powder deposition and melting is repeated until the part of desired shape is achieved. 

The technological parameters used for printing of specimens using the Renishaw AM500E 3D printer were set according to the powder (Renishaw SS 316L) manufacturer’s recommendations to ensure maximum specimen quality with minimal defects. The following process parameters were set: the laser power (P) for powder melting was set at 200 W and the energy of the laser beam (ε) was determined based on Equation (1): 55.9 J.mm^−3^:ε = P/(v × d × t)(1)

Scanning velocity v was 650 mm.s^−1^ and hatching distance d was 110 μm. The thickness of one material layer was t 50 μm. The scanning strategy for individual layers, defining the laser movement path during the powder melting process, was meander. The printing process took place in a protective Argon (Ar) atmosphere.

The dimensions and shapes of the specimens were proposed based on mechanical properties influenced by the printing direction in dimensions (X, Y, Z) of 25 × 100 × 10 mm and 25 × 10 × 100 mm. The specimens of material SS 316L were labeled according to the printing direction as SS 316L A and SS 316L B (Figure 1).

The mechanical properties of the printed material specimens are provided in Table 2 [25].

Material specimens for microstructure evaluation were extracted using AWJ cutting from printed specimens with dimensions of 10 × 10 × 10 mm. The AWJ method does not thermally affect the material. For easier handling of the material specimens, they were cast in cold-cured acrylic resin without pressure. Samples for microstructure observation were meticulously prepared using a multi-step grinding and polishing process. Initially, the samples were ground on a QATM QPOL 250 M2 (ATM Qness GmbH, Mammelzen, Germany) machine using a series of abrasive papers (P320, P600, P1200, P2000, and P4000), employing water to prevent overheating and preserve the integrity of the material. Following grinding, the samples underwent a fine polishing stage on a satin cloth using a diamond paste with a particle size of 0.1 μm. Following the polishing process, the etchant QATM V2A was applied to the samples. The prepared samples were then examined under a Nikon MA100, (Nikon Instruments Inc., Melville, NY, USA) optical microscope at 1000× magnification.

The microstructure of specimens (Figure 2) manufactured using the additive technology of SS 316L steel typically consists of a fine-grained structure with an austenitic matrix. Additionally, the microstructure contains small amounts of secondary phases, such as delta ferrite, which may form due to high temperature gradients and rapid cooling rates during the powder melting process with a laser. In the observed microstructures, characteristic patterns are visible, indicating scan traces of the lasers. In the Z–X and Z–Y directions, the height of individual layers of thickness (t) at 50 µm, corresponding to the set process parameters, can be observed. The presence of delta ferrite can influence material mechanical properties, such as its hardness and toughness. Overall, the microstructure of additively manufactured metallic materials like SS 316L is highly dependent on the set parameters during the printing process. Through parameter optimization and further heat treatment, achieving the desired microstructure and material mechanical properties is possible.

### Preparation and Conduction of the Experiment

The additive manufactured (AM) material was machined in two directions (Figure 3), along the printing layer direction and perpendicular to the printing layer direction, due to the different material properties arising from the preparation technology of the experimental material. To ensure the repeatability of the experiments, 3 sets of experimental specimens for both building directions were prepared. The material machining involved a single pass of the cutting head while maintaining a combination of variable process parameters. Fixed process parameters remained unchanged throughout the whole experiment.

The experiment was conducted using the Water Jet 3015 RT-3D device (Kovostrojservis, spol. s.r.o, Pardubice, Czech Republic) with a table area of 3000 mm × 1500 mm. The machine is equipped with a 45-degree 3D cutting head tiltable in the X and Y direction. Water pressure is generated by a high-pressure pump, PTV Jets 3.8/60. The equipment can generate a maximum water pressure of 415 MPa with a flow rate of 3.8 liters per minute.

The process parameters for the experiment are categorized into fixed and variable. Fixed process parameters are presented in Table 3.

Design of Experiment (DoE) method is a combination of variable process parameters: traverse speed (*v_f_*), abrasive mass flow rate (*m_a_*), and standoff distance (*SoD*). The preliminary experiment was on material AISI 316L with random variation of input variables: traverse speed *v_f_* ranging from 60 mm·min^−1^ to 300 mm·min^−1^; abrasive mass flow rate *m_a_* ranging from 20 g·min^−1^ to 60 g·min^−1^; and a standoff distance *SoD* of 4 mm. The observed parameter was the erosion groove volume (Vd) over a length of 10 mm. The experiment demonstrates the impact of the combination of traverse speed (*v_f_*) and abrasive mass flow rate (*m_a_*) on the erosion groove volume (Vd). The largest observed erosion groove volume, Vd = 2.379 mm^3^, was recorded at a combination of *v_f_* = 60 mm·min^−1^ and *m_a_* = 60 g·min^−1^. The smallest erosion groove volume, Vd = 0.149 mm^3^, was observed at a combination of *v_f_* = 300 mm·min^−1^ and *m_a_* = 20 g·min^−1^. Erosion grooves at traverse speeds above 230 mm·min^−1^ were difficult to evaluate due to the reduced erosion effect. This highlights the significant influence of process parameters on material removal efficiency, with lower traverse speed and higher abrasive mass flow rates producing more substantial erosion, while higher traverse speeds with lower abrasive mass flow rates result in minimal material removal. The difficulty in assessing erosion grooves at higher speeds suggests a threshold beyond which the process becomes less effective. Based on the preliminary experiment, values of input variables for the main experiment were then selected for additive prepared material. The individual levels for the variable process parameters are detailed in Table 4.

In total, 36 combinations of variable process parameters were created with three repetitions (3 × 36) for each material, SS 316L A and SS 316L B. The realization of the experiment is shown in Figure 4. The machined specimens of the material are presented in Figure 5.

## 3. Results and Discussion

Measurement and evaluation of machined specimens were conducted using a 3D digital microscope Keyence VHX 5000, and an optical profilometer Microprof FRT. The parameters of each erosion groove were measured along lines. In each line, the profile of the erosion groove perpendicular to the machining direction was scanned. The main observed parameters of the erosion groove included its shape, removed material area, maximum depth, and width. 

For each erosion groove, 11 lines of groove profiles were created. Profiles of the erosion groove along a vertical cross-section in each line were obtained to measure their parameters (removed material area, maximum depth, and width) (Figure 6). The data acquisition process is illustrated in Figure 7 for a non-transient erosion groove in the SS 316L B material with the set process parameters: traverse speed *v_f_* 60 mm·min^−1^, abrasive mass flow *m_a_* 60 g·min^−1^, and standoff distance *SoD* 4 mm.

From the measured line profiles of the erosion groove, the cross-cut area was selected as the significant observed parameter. The area of the erosion groove represents the removed material area. For material SS 316L A, this area is marked as S_D_ SS 316L A, and for material SS 316L B it is marked as S_D_ SS 316L B. The average values of the removed material areas for the set process parameters and repetitions of the experiment according to the experimental plan are presented in Table 5.

A graphical representation of average areas of erosion grooves (S_D_ SS 316L A and S_D_ SS 316L B) for all combinations of technological parameters is shown in Figure 8. 

From the measured values, it is evident that the significant technological parameter is the traverse speed *v_f_* in combination with the abrasive mass flow rate *m_a_*. The highest values of the average area of erosion groove were recorded at a traverse speed of *v_f_* 60 mm·min^−1^, an abrasive mass flow rate of *m_a_* 60 g·min^−1^, and a standoff distance *SoD* of 4 mm. For material SS 316L A, the highest measured average value of the erosion groove (S_D_ SS 316L A) is 268,092 µm² (Figure 9), and for material SS 316L B, the highest measured average value of the erosion area groove (S_D_ SS 316L B) is 283,538 µm² (Figure 10). 

The lowest values of the average erosion area groove were recorded at a traverse speed of *v_f_* 220 mm·min^−1^ and an abrasive mass flow rate of *m_a_* 20 g·min^−1^. For material SS 316L A, S_D_ SS 316L A is 27,303 µm^2^ (Figure 11), and for SS 316L B (Figure 12), the average erosion area of groove (S_D_ SS 316L B) is 38,615 µm², with standoff distance *SoD* at 5 mm. 

From the graph representing the measured values of the average area of erosion groove, it is evident that the erosive effect decreases with increasing traverse speed *v_f_*. As the abrasive mass flow rate *m_a_* increases in combinations of traverse speeds *v_f_* and standoff distance *SoD*, the erosive effect tends to increase. Comparing the sizes of average area of erosive grooves among the tested materials, it can be observed that the material labeled as SS 316L B exhibits a significantly higher average area of erosion groove in all combinations of variable technological parameters.

### 3.1. ANOVA and Regression Analysis (Average Area of Erosion Groove S_D_ SS 316L A)

Based on ANOVA, it is possible to assess the statistical significance of individual input factors on the observed output variable. The input factors in the form of traverse speed *v_f_*, abrasive mass flow *m_a_*, and standoff distance *SoD* were evaluated for their impact on the observed output value: average area of erosion groove S_D_ SS 316L A, using ANOVA. ANOVA was created for both observed variables (SS 316L A and SS 316L B).

The analysis of variance (ANOVA) (Table 6) reveals significant insights into the regression model’s performance and the individual effects of predictor variables. The overall regression model demonstrates statistical significance (α = 0.05), indicating that at least one predictor variable has a significant impact on the response variable. Further examination of the individual predictors, *v_f_*, *m_a_*, and *SoD* shows that all are highly significant contributors to the model (α = 0.05). Additionally, interactions between variables were explored, with the *v_f_∙m_a_* interaction proving to have a significant effect on the response, while interactions involving *v_f_∙SoD* and *m_a_∙SoD* were not statistically significant. These findings suggest that the combined effects of predictors significantly influence the response, providing valuable insights for future analysis and modeling.

The coefficient of determination R^2^ for the model reached a value of 0.9946, indicating that this model can explain 99.46% of the variability in the model. This indicates that the model has high significance. The influences of the set technological parameters (*v_f_*, *m_a_*, and *S_O_D*) on the observed dependent variable (average area of erosion groove S_D_ SS 316L A), are described by the regression Equation (2):S_D_ SS 316L A = 108538 − 482 *v_f_* + 4192 *m_a_* − 7416 SoD − 16.03 *v_f_* × *m_a_* + 57.0 *v_f_* × SoD − 100 *m_a_* × *SoD*(2)

The Pareto chart of the standardized factors for technological parameters and the specific combinations of the regression model is shown in Figure 13. The critical value for the significance of factors and their interactions is 2.31. The Pareto chart (Figure 13) illustrates the influences of input parameters and their combinations on the observed dependent variable SD SS 316L A. The predictor A, corresponding to the traverse speed *v_f_* with a value of 28.58, has the greatest impact. The second impact with value 11.8 corresponds to predictor B (*m_a_*). Just on the significance boundary is the predictor C, standoff distance *SoD* with a value of 2.51. 

In the following figure (Figure 14), the change in the value of the dependent variable *SD* SS 316L A is observed with respect to the main input parameters, abrasive mass flow *m_a_*, and traverse speed *v_f_*. Based on the graph, as the value of ma increases and the value of *v_f_* decreases, the dependent observed value SD SS 316L A increases. Material removal is primarily influenced by the increase in abrasive mass flow *m_a_* and a decrease in the traverse speed *v_f_* of the AWJ cutting head.

### 3.2. ANOVA and Regression Analysis (Average Area of Erosion Groove S_D_ SS 316L B)

Specimens of SS 316L material, labeled as SS 316L B, were machined perpendicular to the direction of the printing layers due to different mechanical properties. This was done for the purpose of comparing the resulting monitored value of material loss (average value of erosion groove) with specimens labeled as SS 316L A. ANOVA and regression analysis were conducted for the dependent variable, the average value of erosion groove S_D_ SS 316L B.

The analysis of variance (ANOVA) (Table 7), examining the dependent variable S_D_ SS 316L B, provides significant insights into the performance of the regression model and the individual effects of the predictive variables. The overall regression model demonstrates statistical significance (α = 0.05). Further examination of the individual predictors, linear, *v_f_*, *m_a_*, and *SoD*, shows that all contribute to the model with high significance (α = 0.05). The interaction *v_f_∙m_a_* has been identified as a significant factor influencing the explained variable (SD SS 316L B), while interactions involving *v_f_∙SoD* and *m_a_∙SoD* are not statistically significant.

The model’s coefficient of determination R^2^ achieved a value of 0.9950, suggesting model accuracy for 99.50% of the variability in the data. This indicates the model’s high significance. The regression Equation (3) describes the effects of the specified technological parameters (*v_f_*, *m_a_*, and *SoD*) on the observed dependent variable (the average area of erosion groove) for the additive manufactured material SS 316L B.
S_D_ SS 316L B = 107959 − 336 *v_f_* + 3993 *m_a_* − 6625 SoD − 15.41 *v_f_* × *m_a_* + 23 *v_f_* × SoD − 35 *m_a_* × SoD(3)

The Pareto chart (Figure 15) looks similar to the variable S_D_ SS 316L A. The critical value for the significance of factors and their interactions is 2.31. The most significant influence in this model is predictor A corresponding to *v_f_* with a value of 29.28, followed by predictor B (*m_a_*) on 13.78, and the interaction of predictors A (*v_f_*) and B (*m_a_*) is 7.33. Predictor C (*SoD*) is on the boundary of the critical value of significance (2.95). The interactions of predictors BC (*v_f_∙SoD*) and AC (*m_a_∙SoD*) are below the critical value threshold (2.31) and therefore negligible for building the regression model.

In following figure (Figure 16), the change in the value of the dependent variable SD SS 316L B is shown in relation to the primary input parameters: abrasive mass flow *ma* and traverse speed *vf*, standoff distance *SoD* fixed on 4 mm. 

Comparing specimens SS 316L A and SS 316L B, both representing the same material produced by additive manufacturing (AM), in experiments focused on the creation of non-transient erosion grooves under specific technological conditions of traverse speed *v_f_* at levels 60 mm·min^−1^, 140 mm·min^−1^, and 220 mm·min^−1^, abrasive mass flow *m_a_* at levels 20 g·min^−1^, 40 g·min^−1^, and 60 g·min^−1^, and standoff distance *SoD* at 4 mm, 3 mm, and 5 mm, the most significant technological parameters appear to be traverse speed *v_f_*, abrasive mass flow *m_a_*, and their second-order interactions. The technological parameter in the form of standoff distance *SoD* and its variation, with the standard value of 4 mm and a variation of ±1 mm, seems to be a less significant factor in this experiment. From the data obtained in the experiment involving the division of the same material into SS 316L A and SS 316L B, differences in the properties of the 3D printed AM material were observed, resulting from the layering direction during AWJ machining and the formation of erosion grooves.

The AM material prepared by the SLM method exhibits various mechanical properties resulting from its SLM preparation method. The powder material manufacturer also specifies different mechanical properties. For this reason, the SS 316L material was divided into SS 316L A and SS 316L B during the experiments and these were evaluated separately. The study highlights the distinct behavior of AM material when machined using low pressure in AWJ technology. A comparison of the experiments reveals limited relevant sources for comparing experimental results. Most available relevant sources from other authors are focused on high-pressure machining AWJ of conventional material AISI 316L. Other available sources use water pressures of at least 100 MPa or higher traverse speeds above 100 mm·min^−1^ for machining.

The Design of Experiments (DoE) method was applied to determine the influence of input dependent parameters (*v_f_*, *m_a_*, and *SoD*) on the monitored response in the form of average removed material area (S_D_ SS 316L A and S_D_ SS 316L B). Based on the analysis, the most significant technological parameters appear to be traverse speed *v_f_*, abrasive mass flow *m_a_*, and their second-order interactions. Standoff distance *SoD* seems to be a less significant factor in this experiment. When dividing the SS 316L material into SS 316L A and SS 316L B, differences in the values of the monitored variables indicate variations in the properties of the AM material.

Due to the fact that there is a lack of published research dealing with the AWJ machining of AM materials, it is hard to critically compare the obtained results. Low-pressure AWJ machining of titanium alloy was performed by Botko et al. [20]. According to their results, 50 MPa pressure was shown to be appropriate for effective material removal for both casted and additive manufactured materials. A significant influence of traverse speed and abrasives mass flow in combination with 50 MPa pressure on removal characteristics was observed. Holmberg et al. [28] applied different pressures (50–300 MPa) and standoff distances (5–30 mm) and significance of SoD was displayed for low traverse speed. In the presented research, SoD was less significant, probably due to the lower range of limit for SoD values. Considering the results of other studies focused on AWJ milling, Alberdi et al. [6] emphasized the influence of process parameters on the kerf geometry in various metal alloys and found that parameters such as abrasive mass flow and traverse speed significantly affect the milling quality. The results of the presented experiment confirm these findings, particularly the influence of traverse speed and abrasive mass flow on erosion grooves. This suggests that optimization of these parameters is crucial for achieving the desired results in AWJ milling. Fowler et al. [15] investigated AWJ controlled depth milling of the Ti6Al4V alloy and found that increasing the traverse speed reduces the depth of cut, similar to the findings in this experiment where higher traverse speeds led to lower erosion grooves. These findings highlight the importance of traverse speed in controlling the erosion depth and surface quality during AWJ milling. Hashish, M. [16,17] demonstrated the precision of AWJ controlled depth milling techniques, thereby reinforcing the potential of AWJ for precise milling applications. In the presented study, the results show that material removal using AWJ can achieve high precision with properly set process parameters. Escobar-Palafox et al. [7] characterized AWJ pocket milling of the Inconel 718 alloy and emphasized the optimization of process parameters. The findings of this investigation could be used for future studies focused on pocket milling of additive manufactured materials using AWJ. 

The added value of the presented experimental research lies in the testing of material removal for two directions of AM layer building.

## 4. Conclusions

From the obtained results, we can conclude that the area of erosion grooves of materials S_D_ SS 316L A and S_D_ SS 316l B is mainly influenced by the selected input technological conditions, especially traverse speed (*vf*) and abrasive mass flow (*m_a_*). The variation of the standoff distance (*SoD*) parameter proved to be a less significant factor in this experiment. Material SS 316L A exhibits significantly higher values of the evaluated variables (average area of the erosion groove S_D_ SS 316L A). This is attributed to the diverse properties of material SS 316L resulting from its preparation by the SLM method. It was found that with increasing values of the abrasive mass flow (*m_a_*) and traverse speed (*v_f_*), material loss and erosion effects increase.
-The highest average values of the erosion groove were recorded at traverse speed *v_f_* 60 mm·min^−1^, abrasive mass flow rate *m_a_* 60 g·min^−1^, and standoff distance SoD 4 mm. For material SS 316L A, the highest measured average value of the area of erosion groove was S_D_ SS 316L A 268,092 µm^2^. -For material SS 316L B, the highest measured average value of the area of erosion groove was S_D_ SS 316L B 283,538 µm^2^. The lowest average values of the erosion groove were recorded at a traverse speed of 200 mm·min^−1^ and an abrasive mass flow rate *m_a_* of 20 g·min^−1^ with standoff distance *SoD* 5 mm. -For specimens of material SS 316L A, S_D_ SS 316L A 27,303 µm^2^, and for SS 316L B, the average area of the erosion groove was S_D_ SS 316L B 38,615 µm^2^.


The potential of the method lies in the proper adjustment of input process parameters and the combination of multiple passes of the AWJ cutting head, demonstrating the potential of AWJ milling technology. By utilizing low water pressure at 50 MPa, the process itself becomes more efficient and less energy-demanding. This presents an opportunity for researchers to harness the potential of this method in practical applications. By optimizing the process parameters and exploring its capabilities further, AWJ milling can become a viable and efficient method for machining AM materials in various industries.

## Figures and Tables

**Figure 1 materials-17-02964-f001:**
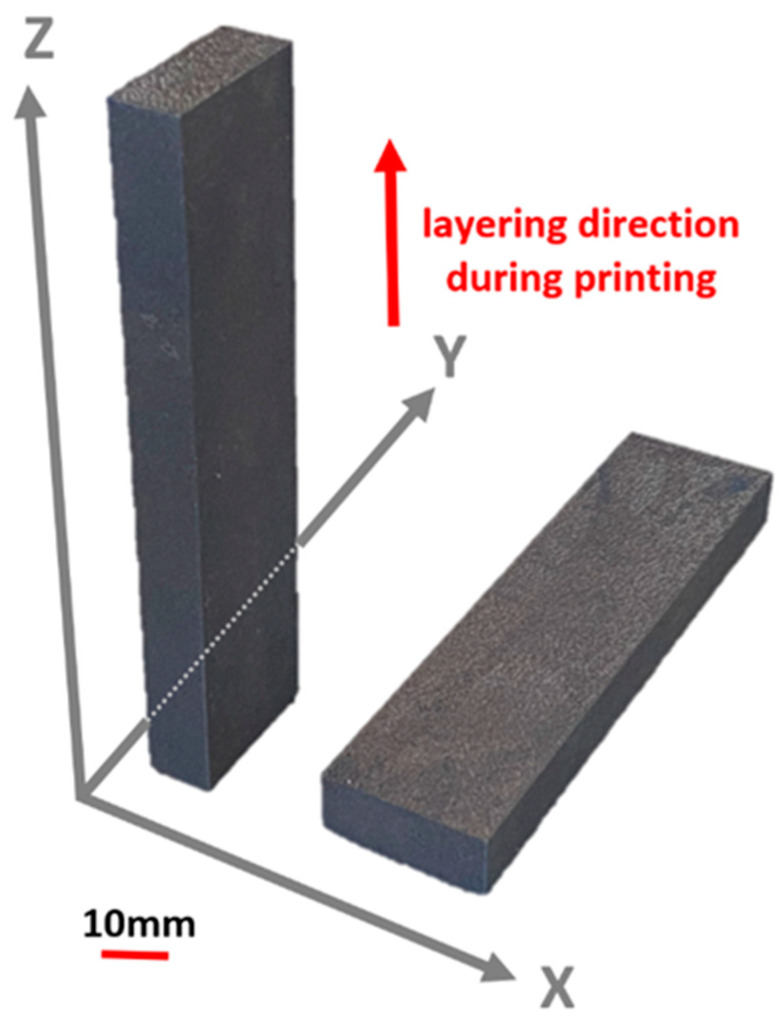
Specimens of SS 316L material, SS 316L A on the left, SS 316L B on the right.

**Figure 2 materials-17-02964-f002:**
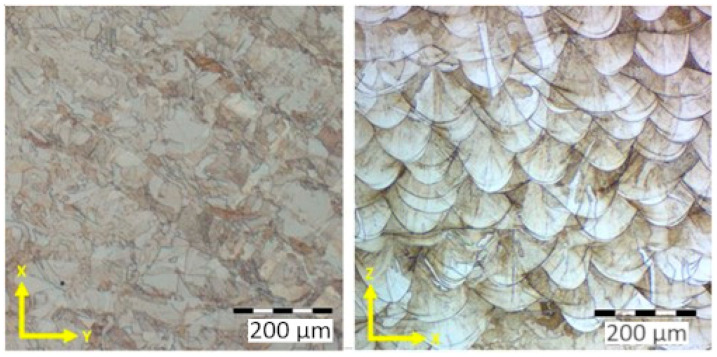
Microstructure of AM specimens (Z-direction of print layering).

**Figure 3 materials-17-02964-f003:**
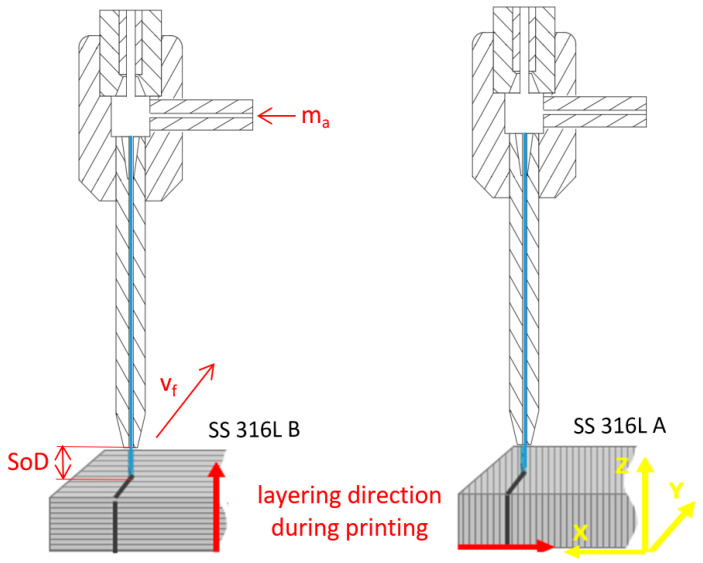
Experimental AWJ machining of AM SS 316L specimens.

**Figure 4 materials-17-02964-f004:**
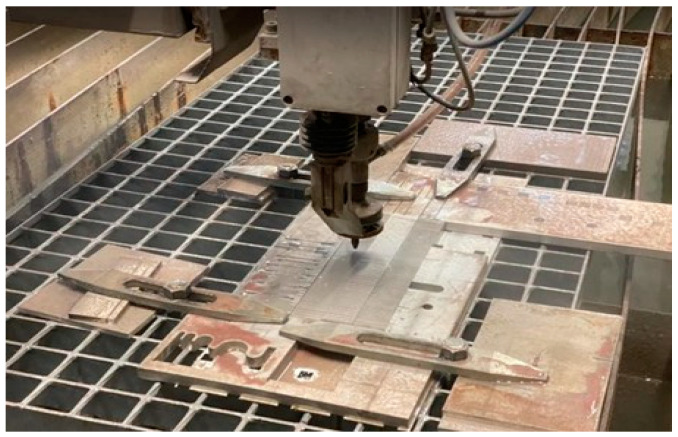
Realization of experiment.

**Figure 5 materials-17-02964-f005:**
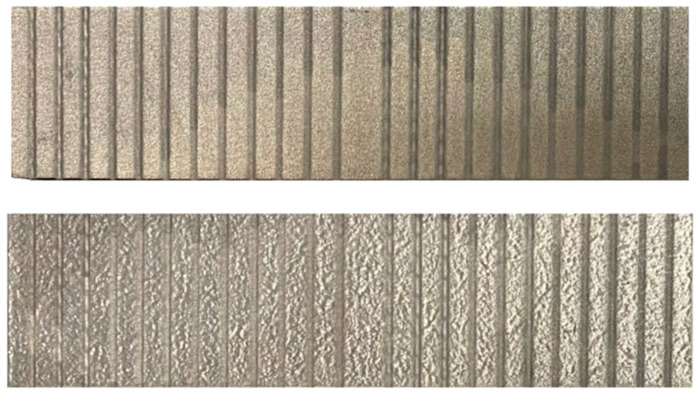
Machined specimens of materials SS 316L A and SS 316L B.

**Figure 6 materials-17-02964-f006:**
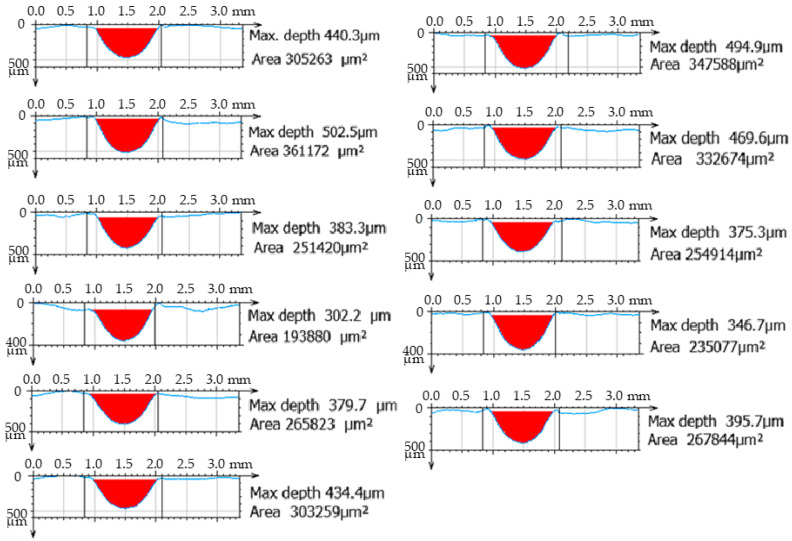
Cross-sections of the erosion groove, *v_f_* 60 mm·min^−1^, *m_a_* 60 g·min^−1^, *SoD* 4 mm, and material SS 316L B.

**Figure 7 materials-17-02964-f007:**
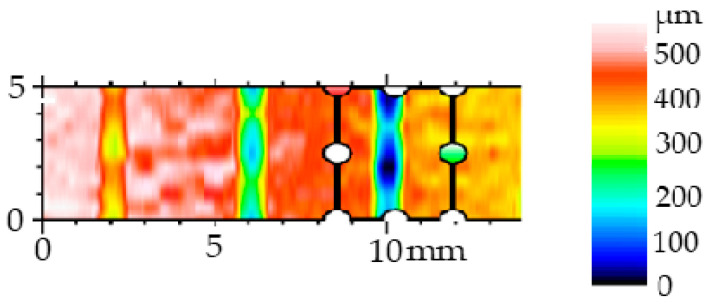
Visualization of the erosion groove and lines for measuring the profile of the erosion groove, *v_f_* 60 mm·min^−1^, *m_a_* 60 g·min^−1^, *SoD* 4 mm, and material SS 316L B.

**Figure 8 materials-17-02964-f008:**
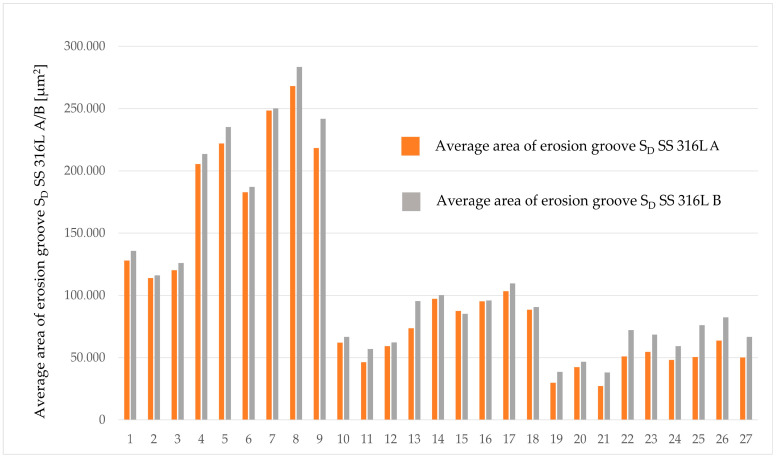
Average areas of erosion grooves.

**Figure 9 materials-17-02964-f009:**
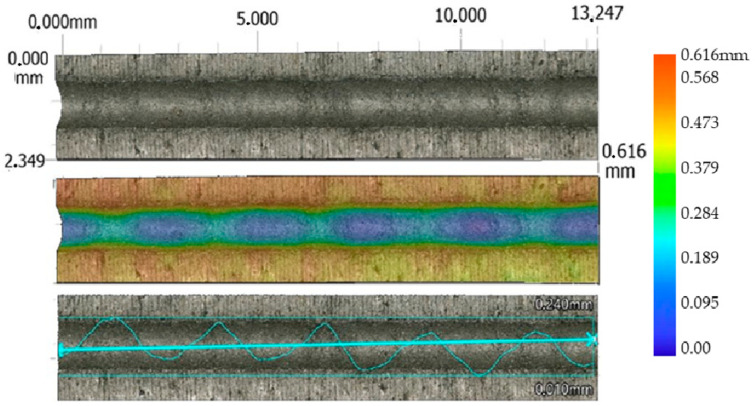
Profile of AWJ-created erosion groove, material SS 316L A (*v_f_* 60 mm·min^−1^, *m_a_* 60 g·min^−1^, and *SoD* 4 mm).

**Figure 10 materials-17-02964-f010:**
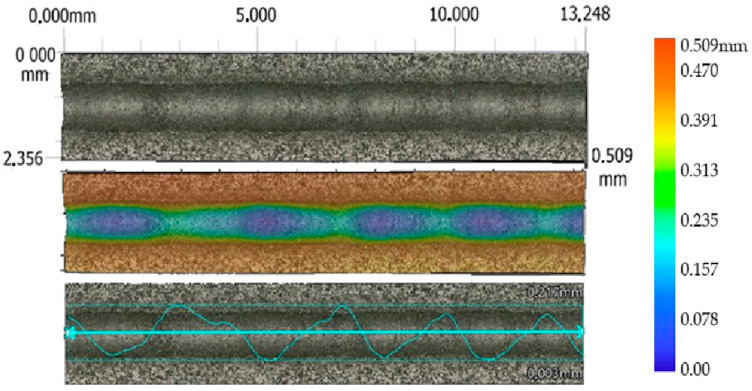
Profile of AWJ-created erosion groove, material SS 316L B (*v_f_* 60 mm·min^−1^, *m_a_* 60 g·min^−1^, and *SoD* 4 mm).

**Figure 11 materials-17-02964-f011:**
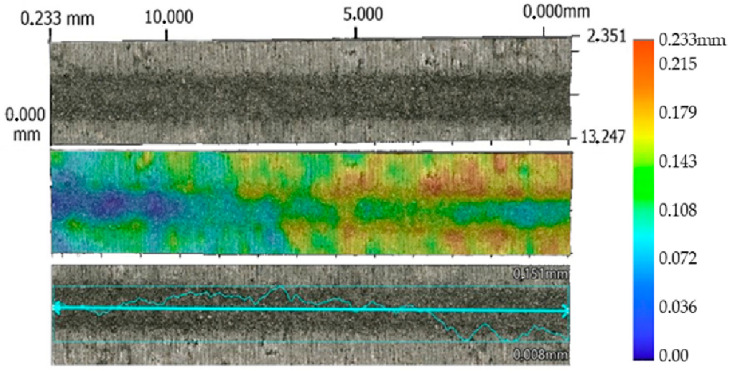
Profile of AWJ-created erosion groove, material SS 316L A (*v_f_* 220 mm·min^−1^, *m_a_* 20 g·min^−1^, and *SoD* 5 mm).

**Figure 12 materials-17-02964-f012:**
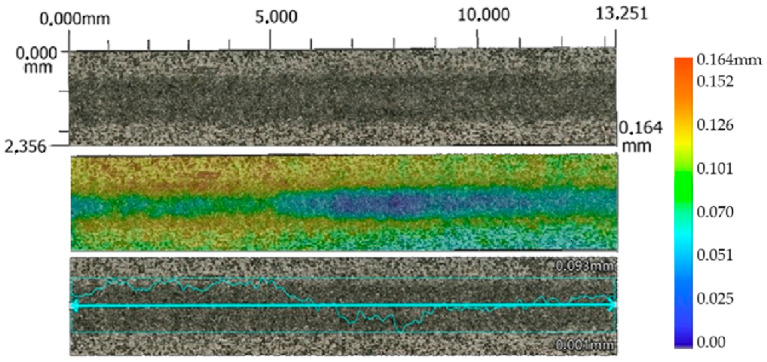
Profile of AWJ-created erosion groove, material SS 316L B (*v_f_* 220 mm·min^−1^, *m_a_* 20 g·min^−1^, and *SoD* 5mm).

**Figure 13 materials-17-02964-f013:**
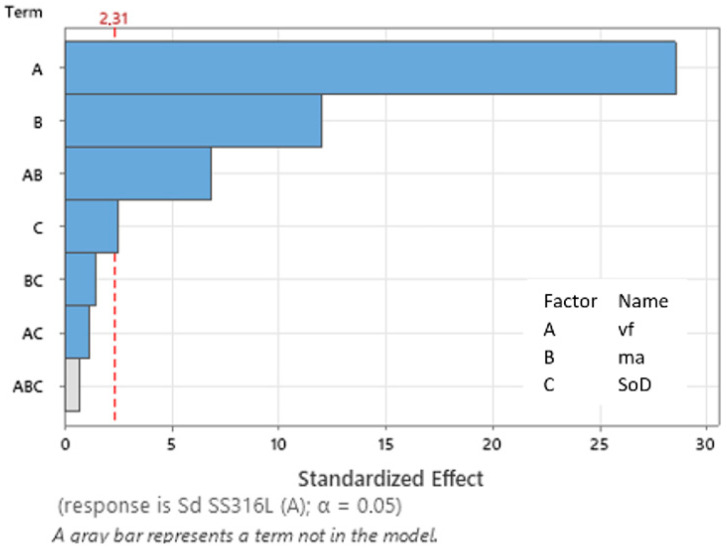
Pareto chart of the standardized effect (S_D_ SS 316L A).

**Figure 14 materials-17-02964-f014:**
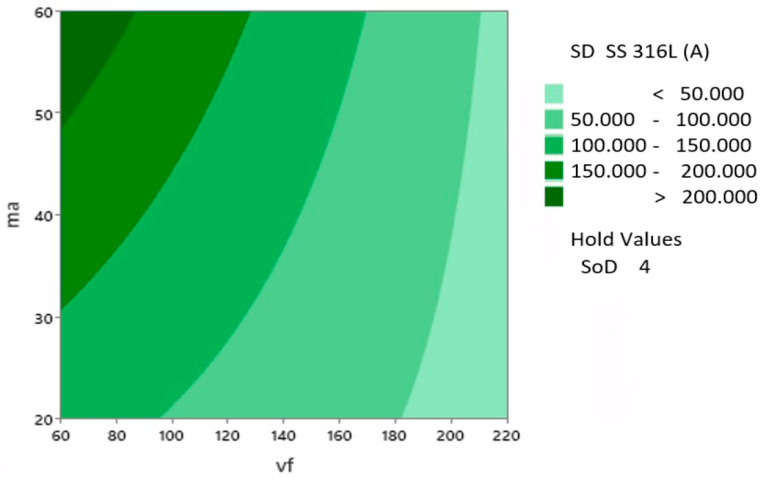
Contour plot of S_D_ SS 316L A vs. *m_a_*, *v_f_*, *SoD* hold 4 mm.

**Figure 15 materials-17-02964-f015:**
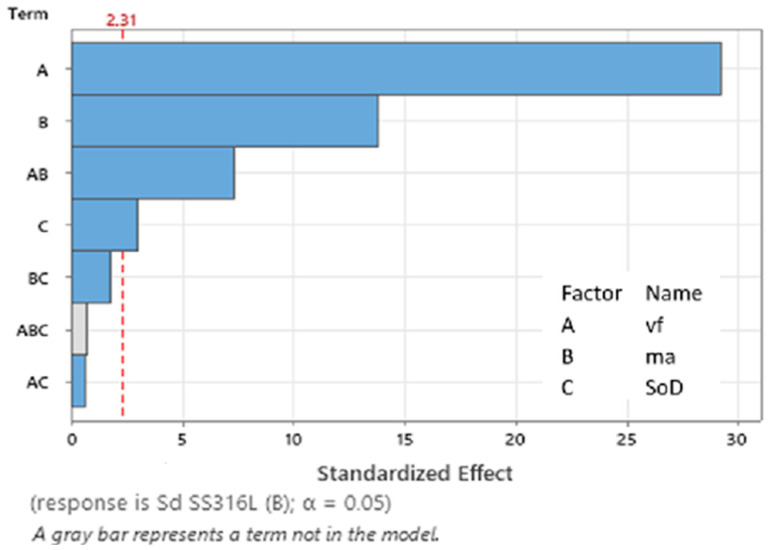
Pareto chart of the standardized effect (S_D_ SS 316L B).

**Figure 16 materials-17-02964-f016:**
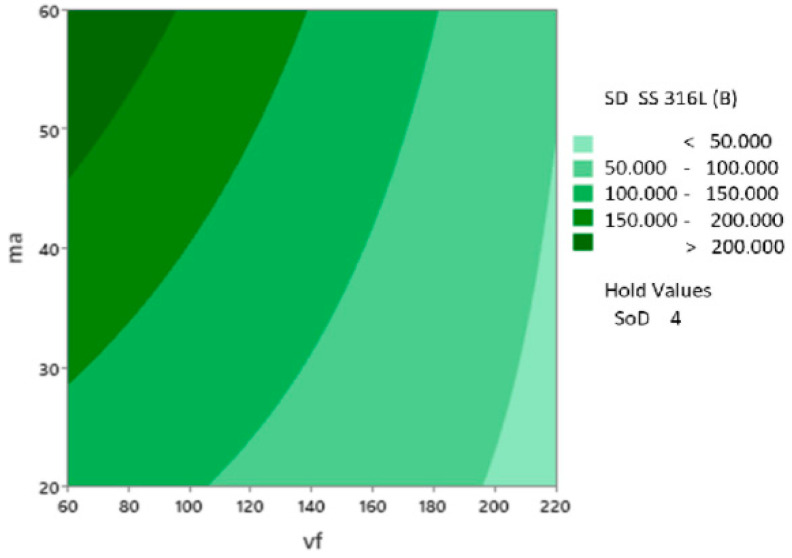
Contour plot of S_D_ SS 316L A vs. *m_a_*, *v_f_*, *SoD* fixed on 4 mm.

**Table 1 materials-17-02964-t001:** Renishaw SS 316L-0407 powder composition [24].

Element	Cr	Ni	Mo	Mn	Si	N	O	P	C	S	Fe
Mass (%)	16–18	10–14	2–3	≤2	≤1	≤0.1	≤0.1	≤0.045	≤0.03	≤0.03	Rest.

**Table 2 materials-17-02964-t002:** Mechanical properties of additively manufactured components, SS 316L [25].

Orientation	Upper Tensile Strength ^1^	Yield Strength ^1^	Elongation at Break ^1^	Modulus of Elasticity ^1^	Hardness (Vickers) ^2^
Horizontal direction (XY)	676 ± 2 MPa	547 ± 3 MPa	43 ± 2%	197 ± 4 GPa	198 ± 8 HV0.5
Vertical direction (Z)	624 ± 17 MPa	494 ± 14 MPa	35 ± 8%	190 ± 10 GPa	208 ± 6 HV0.5

As built: ^1^ Tested at ambient temperature by Nadcap- and UKAS-accredited independent laboratory. Test ASTM E8 [26]. Machined prior to testing. ^2^ Tested according to ASTM E384-11 [27], after polishing.

**Table 3 materials-17-02964-t003:** Fixed process parameters of the experiment.

Water Pressure *p*	Nozzle Diameter dv	Diameter of Focusing Tube df	Length of Focusing Tube dL	Tilt Angle of Cutting Head γ
50 MPa	0.33 mm	1.02 mm	76.2 mm	90°

**Table 4 materials-17-02964-t004:** Levels of variable process parameters.

Level	Traverse Speed *v_f_*	Abrasive Mass Flow *m_a_*	Standoff Distance *SoD*
1	60 mm·min^−1^	20 g·min^−1^	3 mm
2	140 mm·min^−1^	40 g·min^−1^	4 mm
3	220 mm·min^−1^	60 g·min^−1^	5 mm

**Table 5 materials-17-02964-t005:** Average values of removed material areas.

	Process Parameters			Average Area of Erosion Groove	
	Traverse Speed *v_f_* [mm·min^−1^]	Abrasive Mass Flow *m_a_* [g·min^−1^]	Standoff Distance *SoD* [mm]	S_D_ SS 316L A [µm^2^]	S_D_ SS 316L B [µm^2^]
1	60	20	3	127,992	135,761
2	60	20	4	113,970	116,181
3	60	20	5	120,278	126,079
4	60	40	3	205,556	213,629
5	60	40	4	221,990	235,206
6	60	40	5	182,891	187,181
7	60	60	3	248,444	250,309
8	60	60	4	268,092	283,538
9	60	60	5	218,418	241,875
10	140	20	3	62,153	66,663
11	140	20	4	46,431	57,003
12	140	20	5	59,306	62,237
13	140	40	3	73,695	95,421
14	140	40	4	97,359	100,296
15	140	40	5	87,468	85,199
16	140	60	3	95,349	95,985
17	140	60	4	103,309	109,608
18	140	60	5	88,549	90,729
19	220	20	3	29,907	38,615
20	220	20	4	42,399	46,743
21	220	20	5	27,303	38,127
22	220	40	3	50,942	72,112
23	220	40	4	54,720	68,516
24	220	40	5	48,258	59,235
25	220	60	3	50,552	76,072
26	220	60	4	63,774	82,445
27	220	60	5	50,168	66,732

**Table 6 materials-17-02964-t006:** Analysis of variance (ANOVA) for S_D_ SS 316L A.

Source	DF	Adj SS	Adj MS	F-Value	*p*-Value
Regression	18	131031000000	7279491175	81.23	0.00
Linear	6	120563000000	20093804546	224.23	0.00
*v_f_*	2	101427000000	50713437871	565.91	0.00
*m_a_*	2	18205076623	9102538312	101.58	0.00
*SoD*	2	930874909	465437454	5.19	0.04
2-Way Interactions	12	10468013876	872334490	9.73	0.00
*v_f_∙m_a_*	4	9191262538	2297815635	25.64	0.00
*v_f_∙SoD*	4	567226430	141806607	1.58	0.27
*m_a_∙SoD*	4	709524908	177381227	1.98	0.19
Error	8	716908707	89613588		
Total	26	131748000000			

**Table 7 materials-17-02964-t007:** Analysis of variance (ANOVA) for S_D_ SS 316L B.

Source	DF	Adj SS	Adj MS	F-Value	*p*-Value
Regression	18	131463000000	7303519548	88.67	0.00
Linear	6	120748000000	20124643121	244.33	0.00
*v_f_*	2	97794694671	48897347335	593.67	0.00
*m_a_*	2	21811494758	10905747379	132.41	0.00
*SoD*	2	1141669298	570834649	6.93	0.02
2-Way Interactions	12	10715493131	892957761	10.84	0.00
*v_f_∙m_a_*	4	9581438798	2395359700	29.08	0.00
*v_f_∙SoD*	4	264743529	66185882	0.80	0.56
*m_a_∙SoD*	4	869310804	217327701	2.64	0.11
Error	8	658920167	82365021		
Total	26	132122000000			

## Data Availability

The original contributions presented in the study are included in the article, further inquiries can be directed to the corresponding author.

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
