# Peer review of "Influence of AWJ Process Parameters on Erosion Groove Formation in Additively Manufactured Stainless Steel"

_materials, 2024, doi:10.3390/ma17122964_

Round 1

Reviewer 1 Report

Comments and Suggestions for Authors

- Abbreviations are not defined first. Please correct it in Abstract.

- Reference in brackets should be before dot.

- Please describe usage in praxe and also industry in the last paragraph in Introduction.

- Elements in Table 1 are in wt. % or at. %?

- Fig. 1 and 5 should have white background.

- There is not description, how was microstructure in Fig. 2 visualized, what kind of methalographic procedures were used. Also there is not list of used microscopes and other equipment.

- Text in Figs. 6 and 7 should have the same size like other text in manuscript.

- Figs. 10-12 should have white background.

- Shorten conclusions and create list with bullets.

- Please create discussion in normal way. Compare your results with results in other studies. Discussion doesnt have to be separate chapter. Maybe will be better to add rich discussion to chapter 3. and create Results and discussion.

Author Response

We would like to thank the reviewer for valuable comments and suggestions to improve our manuscript.

- Abbreviations are not defined first. Please correct it in Abstract.

- Corected

- Reference in brackets should be before dot.

- Corected

- Please describe usage in praxe and also industry in the last paragraph in Introduction.

              -Added to the article.

- Elements in Table 1 are in wt. % or at. %?

- Corected

- Fig. 1 and 5 should have white background.

- Corected

- There is not description, how was microstructure in Fig. 2 visualized, what kind of methalographic procedures were used. Also there is not list of used microscopes and other equipment.

-Added to the article.

- Text in Figs. 6 and 7 should have the same size like other text in manuscript.

- Corected

- Figs. 10-12 should have white background.

- Corected

- Shorten conclusions and create list with bullets.

- Corected

- Please create discussion in normal way. Compare your results with results in other studies. Discussion doesnt have to be separate chapter. Maybe will be better to add rich discussion to chapter 3. and create Results and discussion.

- Corected and added to article.

Reviewer 2 Report

Comments and Suggestions for Authors

The abstract is OK, even was better indicating some impact and more industrial background

Please avoid large block citation “[1-6].” Max 3 ref together are enough

Line 63 requires a reference

I see that you claim some novelty through no study in this aspect, however it will be good to see also the scientific novelty

Figure 1 requires a scale bar

The parameters selected in Table 3 have any physical meaning in respect to industrial process ?

How many sample for each type of trial were used in order to ensure the repeatability

The legend of Figure 6 left hand side is poor /better quality is required

From Figure 8 cannot be expressed “seems to be the traverse speed vf in combination with the abrasive mass flow rate ma” as there is limited variance – otherwise the AM have generally much large variance

Discussion section is very superficial. It truly should discuss your results against literature data

Some more recent literature is required

Comments on the Quality of English Language

Extensive editing of English language required

Author Response

We would like to thank the reviewer for valuable comments and suggestions to improve our manuscript.

-The abstract is OK, even was better indicating some impact and more industrial background

-Added to the article.

-Please avoid large block citation “[1-6].” Max 3 ref together are enough

- Corected

-Line 63 requires a reference

- Corected

-I see that you claim some novelty through no study in this aspect, however it will be good to see also the scientific novelty

-The scientific novelty lies in the creation of erosion grooves in materials such as stainless steel 316L. Surface modification is crucial for subsequent material processing, such as welding. For AM materials prepared by the SLM method, these materials exhibit different properties due to the directional influence of layering during printing.

-Figure 1 requires a scale bar

              -Added

-The parameters selected in Table 3 have any physical meaning in respect to industrial process ?

-The change in parameters affects specific industrial applications. The parameters in Table 3 are standardly used in the industry. The dimensions of the focusing tube depend on the manufacturer and the replacement parts they offer. The PTV Jets 3.8/60 pump used can operate in modes of 50 MPa and 415 MPa.

-How many sample for each type of trial were used in order to ensure the repeatability.

              - To ensure of repeatability of experiments 3 sets of experimental specimens for both building direction were prepared (added to the article).

-The legend of Figure 6 left hand side is poor /better quality is required

              -Corected

-From Figure 8 cannot be expressed “seems to be the traverse speed vf in combination with the abrasive mass flow rate ma” as there is limited variance – otherwise the AM have generally much large variance

              Research was focused on AWJ machining of AM products in two building directions. From the machining point of view these are most significant machining parameters. Variability of AM products properties was not the aim of the presented research. Experimental specimens were produced using optimal parameters of AM process for presented material according to the producer.

-Discussion section is very superficial. It truly should discuss your results against literature data

-Comparison with the works of other researchers is difficult because, according to available sources, no one has paid attention to AM materials.

-Some more recent literature is required

-There are few relevant sources in the field of AM materials and low-pressure AWJ for comparison. Other relevant sources have been cited and added to the article.

Reviewer 3 Report

Comments and Suggestions for Authors

The issues discussed in this work are extremely important from the point of view of modern industry. The experiment is extensive. I believe that the experimental plan lacks reference samples for the steels analysed produced by traditional casting and rolling methods. This approach to the topic would allow the results obtained for the tested sinters to be compared to a reference point.

The second doubt concerns the meaning of the experiment itself: what actual abrasive wear conditions should the proposed experiment simulate?

Despite these comments, I consider the work valuable and recommend its publication in the journal Materials.

Author Response

We would like to thank the reviewer for valuable comments and suggestions to improve our manuscript.

-The issues discussed in this work are extremely important from the point of view of modern industry. The experiment is extensive. I believe that the experimental plan lacks reference samples for the steels analysed produced by traditional casting and rolling methods. This approach to the topic would allow the results obtained for the tested sinters to be compared to a reference point.

-The preliminary experiment was conducted on conventional steel AISI 316L. The influence of process parameters (traverse speed, abrasive mass flow, SoD=4mm) on material removal volume was monitored. The results of the experiment are included and described in the article.

-The second doubt concerns the meaning of the experiment itself: what actual abrasive wear conditions should the proposed experiment simulate?

-The aim of the experiment is to monitor the input parameters of the abrasive wear process encountered during machining with abrasive water jet (AWJ) of additively manufactured stainless steel SS316L. These materials exhibit different mechanical properties due to the printing direction. By changing the process parameters and observing their effects on erosion grooves, the study aims to optimize AWJ machining processes, minimize abrasive wear, and increase machining efficiency.-

Despite these comments, I consider the work valuable and recommend its publication in the journal Materials.

Round 2

Reviewer 1 Report

Comments and Suggestions for Authors

Discussion was not added. Please compare your results with results in other studies. You added only two 22 and 28. This is not enough.

Author Response

We extend our gratitude to the reviewer for their valuable feedback and suggestions aimed at enhancing our manuscript.

-Discussion was not added. Please compare your results with results in other studies. You added only two 22 and 28. This is not enough.

- Added. The results were compared with other studies.

Reviewer 2 Report

Comments and Suggestions for Authors

.

Comments on the Quality of English Language

.

Author Response

I would like to thank the reviewer for the review. English language in the manuscript was corrected.